# Secondary Prevention of Diabetes Type 1 with Oral Calcitriol and Analogs, the PRECAL Study

**DOI:** 10.3390/children10050862

**Published:** 2023-05-11

**Authors:** Dimitrios T. Papadimitriou, Eleni Dermitzaki, Panagiotis Christopoulos, Maria Papagianni, Kleanthis Kleanthous, Chrysanthi Marakaki, Anastasios Papadimitriou, George Mastorakos

**Affiliations:** 1Second Department of Obstetrics and Gynecology, Aretaieion University Hospital, National and Kapodistrian University of Athens, 11527 Athens, Greece; 2Department of Pediatric-Adolescent Endocrinology and Diabetes, Athens Medical Center, 15125 Marousi, Greece; 3Department of Nutrition and Dietetics, University of Thessaly, 42132 Trikala, Greece; 4Unit of Endocrinology, Diabetes and Metabolism, Third Department of Pediatrics, Aristotle University of Thessaloniki, Hippokrateion Hospital of Thessaloniki, 54642 Thessaloniki, Greece; 5Pediatric Endocrinology Unit, Attikon University Hospital, National and Kapodistrian University of Athens, 12462 Haidari, Greece

**Keywords:** prediabetes, type 1 diabetes, prevention, vitamin D, calcitriol, calcitriol analogs, paricalcitol, autoimmunity, β-cell, HLA, type 1 diabetes autoantibodies

## Abstract

Screening for Type 1 Diabetes (T1D, incidence 1:300) with T1D autoantibodies (T1Ab) at ages 2 and 6, while sensitive, lacks a preventive strategy. Cholecalciferol 2000 IU daily since birth reduced T1D by 80% at 1 year. T1D-associated T1Ab negativized within 0.6 years with oral calcitriol in 12 children. To further investigate secondary prevention of T1D with calcitriol and its less calcemic analog, paricalcitol, we initiated a prospective interventional non-randomized clinical trial, the PRECAL study (ISRCTN17354692). In total, 50 high-risk children were included: 44 were positive for T1Ab, and 6 had predisposing for T1D HLA genotypes. Nine T1Ab+ patients had variable impaired glucose tolerance (IGT), four had pre-T1D (3 T1Ab+, 1 HLA+), nine had T1Ab+ new-onset T1D not requiring insulin at diagnosis. T1Ab, thyroid/anti-transglutaminase Abs, glucose/calcium metabolism were determined prior and q3–6 months on calcitriol, 0.05 mcg/Kg/day, or paricalcitol 1–4 mcg × 1–3 times/day p.o. while on cholecalciferol repletion. Available data on 42 (7 dropouts, 1 follow-up < 3 months) patients included: all 26 without pre-T1D/T1D followed for 3.06 (0.5–10) years negativized T1Ab (15 +IAA, 3 IA2, 4 ICA, 2 +GAD, 1 +IAA/+GAD, 1 +ICA/+GAD) within 0.57 (0.32–1.3) years or did not develop to T1D (5 +HLA, follow-up 3 (1–4) years). From four pre-T1D cases, one negativized T1Ab (follow-up 1 year), one +HLA did not progress to T1D (follow-up 3.3 years) and two +T1Ab patients developed T1D in 6 months/3 years. Three out of nine T1D cases progressed immediately to overt disease, six underwent complete remission for 1 year (1 month–2 years). Five +T1Ab patients relapsed and negativized again after resuming therapy. Four (aged <3 years) negativized anti-TPO/TG, and two anti-transglutaminase-IgA. Eight presented mild hypercalciuria/hypercalcemia, resolving with dose titration/discontinuation. Secondary prevention of T1D with calcitriol and paricalcitol seems possible and reasonably safe, if started soon enough after seroconversion.

## 1. Introduction

### 1.1. A Window of Opportunity to Prevent and Revert Seroconversion towards T1D Autoantibodies, and Intercept the Development of Multiple T1D Autoantibodies

Type 1 Diabetes Mellitus (T1D) is the final consequence of β-cell autoimmunity. While islet autoimmunity is an essential component in the pathogenesis of T1D, β-cell stress provokes an immune attack as well [1], even if the primary risk factor for the development of T1D autoantibodies (T1Ab) is genetic in origin occurring mainly in individuals carrying specific haplotypes of human leukocyte antigen (HLA), which regulate the immune responses and recognition of self versus non-self cells [2,3]. While genetic susceptibility predisposes an individual to diabetes, environmental factors seem to be important in triggering the development of autoimmunity and T1D onset [4]. Classically, T1D is classified into three stages: Stage 1 is characterized by the presence of T1Ab and the absence of dysglycemia, Stage 2 is characterized by the presence of both T1Ab and dysglycemia, and Stage 3 corresponds to symptomatic T1D [5]. Attempts to stage autoimmune diabetes are useful when enrolling individuals in secondary prevention trials. Insel and colleagues [6] proposed three different stages in T1D pathogenesis. In Stage 1, two or more T1Ab are present, but the carrier is normoglycemic. The 5-year and 10-year risks of symptomatic disease are approximately 44% and 70%, respectively, and the lifetime risk approaches 100%. In Stage 2, glucose intolerance or dysglycemia is developed due to loss of functional β-cell mass [7]. The 5-year risk of symptomatic disease at this stage is approximately 75%, and the lifetime risk approaches 100%. In Stage 3, clinical symptoms are present. However, even if only one T1Ab is present, both the age of its appearance and specifically the appearance of insulin auto-antibodies (IAA) are major determinants of the age of Stage 3 T1D onset. IAA are almost unanimously the first to appear especially in T1D children aged <12 years, detected already by the age of 18 months, antedating other T1Ab by a mean of 2.3 (0.3–7.2) years [8]. Therefore, the appearance of T1Ab is the earliest sign of autoimmunity directed towards the pancreatic islet β-cells. Non-maternal T1Ab may be detected as early as at 3 months of age with increasing numbers at 6 months, suggesting a narrow window during which seroconversion appears [9]. The peak incidence of the appearance of the first T1Ab is at the age of 9–24 months for IAA and about 36 months of age for anti-glutamic acid decarboxylase (GAD). Tyrosine phosphatase-related islet antigen 2 antibodies (IA2) and zinc transporter 8 antibody (ZnT8) rarely appear as a first T1Ab and tend to occur later. Of course, the risk of T1D is strongly associated with the number of positive T1Ab. Approximately 70% of cases with seroconversion of multiple T1Ab progress to T1D within 10 years and almost all do so at 15 years compared to 15% with a single T1Ab, the duration from the first seroconversion until T1D appearance varying from a few months to decades [10]. Younger age at the appearance of islet autoimmunity, progression from single to multiple T1Ab and higher IAA titers (but not GAD or IA2) may predict a faster clinical onset according to the Diabetes Autoimmunity Study in the Young (DAISY) cohort [11]. Conversely, in The Environmental Determinants of Diabetes in the Young (TEDDY) study, higher IAA and IA2 titers, but not GAD titers, were linked to an increased T1D incidence in the 5 years following the detection of the first T1Ab [12]. Autoantibody titers do not always increase closer to the time of diabetes onset [13], and up to 33 and 57% inverse seroconversion rates were reported within 6 years for GAD and IA2, respectively, in a non-diabetic sample population aged 3–18 years [14]. 

HLA remains, by far, the strongest predictor of T1D risk [15]. Data from the Finnish population-based Type 1 Diabetes Prediction and Prevention (DIPP) study showed that overall, 3.5% of HLA-positive children will have developed T1D by 15.5 years and 27.7% will have developed islet autoimmunity by 7.4 years of age [16]. While IAA and ZnT8 are the first autoantibodies to be detected in childhood and even before the age of 2 years, islet-cell antibodies (ICA) tend to increase towards puberty. Specifically, children positive to IAA eventually develop GAD and IA2. While inverse IAA seroconversions occurred frequently (49.3% turned negative), this did not revoke T1D risk but meant a prolonged delay from seroconversion to diagnosis compared to persistent IAA (8.2 vs. 3.4 years). As age is inversely related to T1D risk in those with multiple T1Ab [16], and the possibility of progression to T1D increases almost fivefold with multiple positive T1Ab [10], it is crucial to develop interventions to timely prevent the initial seroconversion, revert it if possible by achieving reversion of seroconversion by negativizing the positive T1Ab, and intercept the development of multiple T1Ab, well before any progress to pre-T1D occurs.

### 1.2. The Role of Vitamin D

Vitamin D seems to play a significant role in the incidence and progression as well as in the metabolic control and complications of T1D [17]. Beyond the descriptive association between hypovitaminosis D and T1D [18], vitamin D and its nuclear receptor VDR seem to have a combined role in the development of islet autoimmunity in children at increased genetic risk for T1D [19]. Both pancreatic insulin-producing β-cells and immune system cells express the VDR and vitamin D-binding protein (DBP), polymorphisms in which are involved in the pathogenesis of T1D in genetically susceptible individuals [20]. Higher maternal DBP levels at delivery as well as increased 25(OH)D levels at birth may decrease T1D risk, depending on VDR genotype [21]. Cholecalciferol not only enhances the number of Treg cells, but it also improves their suppressive capacity [22]. Maintenance of optimal circulating 25(OH)D with daily cholecalciferol supplementation in children with T1D may delay the development of absolute or near-absolute c-peptide deficiency, exerting a protective role against long-term complications preventing micro- and macrovascular complications related to chronic hyperglycemia and improving endothelial function [23]. The immunomodulatory and β-cell conserving properties of vitamin D, the active hormone calcitriol and its analogs, suggest a therapeutic potential for vitamin D in autoimmune diseases and notably T1D [24]. While vitamin D supplementation may be beneficial in inflammation, infections, malignancies, cardiovascular diseases and autoimmune disorders, to obtain therapeutic immunomodulating effects in vivo requires supra-physiological doses of 1,25(OH)2D3, which are associated with the undesired risk of hypercalciuria and hypercalcemia [25]. Consequently, novel therapeutic strategies point toward vitamin D analogs that do not induce hypercalcemia, which may exert considerable immunomodulatory activity at non-hypercalcemic dosages and may have therapeutic potential for immune disorders or transplant rejection. The rationale of using less calcemic calcitriol analogs such as paricalcitol, a synthetic biologically active vitamin D calcitriol analog (three times less potent than calcitriol but with ten times fewer calcemic effects), lies in the ability to use clearly higher pharmaceutical doses than those that could be tolerated by using calcitriol due to its calcemic effects [26,27,28]. 

Calcitriol (1,25-Dihydroxyvitamin D), the active metabolite of vitamin D, is literally a hormone and the physiological ligand of the VDR, a nuclear hormone receptor. Apart from its role in maintaining calcium and phosphate homeostasis, in addition to the gut and bone, VDR is present in virtually all tissues, and its natural and synthetic ligands are increasingly recognized for their non-calcemic actions, such as potent antiproliferative, prodifferentiative, and immunomodulatory activities providing a rationale for using VDR ligands for various therapeutic indications, such as rheumatoid arthritis, psoriasis, actinic keratosis, atopic dermatitis [29], autoimmune alopecia [30], osteoporosis, autoimmune diseases (T1D, lupus erythematosus, and multiple sclerosis), as well as several cancers (prostate, colon, breast, leukemia, head, and neck) [31]. Pharmacologic doses of calcitriol have been used to prevent insulitis and T1D in non-obese diabetic (NOD) mice, as well as in other models of T1D, possibly via immune modulation as well as direct effects on β-cell function [32]. 

### 1.3. Primary Prevention of Type 1 Diabetes with Vitamin D

With all of the compelling evidence discussed above, one should wonder what would be the landscape of human health and explicitly that of autoimmunity, cancer, infectious diseases and the metabolic syndrome [33] if optimal vitamin D milieu was assured from the beginning, meaning no other than the conception itself. Indeed, 2000 IU cholecalciferol daily in a large birth cohort study reduced the risk of T1D by 80% in 1 year [34]. T1D hits about 1:300 children, with rising incidence affecting more and more younger children as Finnish data showed [35]. However, in Finland, T1D has plateaued and decreased after an increase in population 25(OH)D concentrations just above 30 ng/mL, the result of the country’s decision for fortification of all dairy products with vitamin D [36]. Population screening at ages 2 and 6 years with T1Ab has been proven sensitive [37]. However, while potential treatments to prevent or delay progress to T1D are currently in development with teplizumab-mzwv very recently receiving FDA approval “https://www.fda.gov/news-events/press-announcements/fda-approves-first-drug-can-delay-onset-type-1-diabetes (accessed on 15 March 2023)”, a population-based cost-effective feasible and safe preventive strategy aiming at the seroconversion window before any progress to Stages 1 and 2 of T1D is still lacking. A global primary prevention strategy or just in those carrying high-risk HLA epitopes could possibly rely on timely and optimal vitamin D supplementation [38].

### 1.4. Secondary Prevention of Type 1 Diabetes with Calcitriol and Paricalcitol

Ten years ago, a small pilot prospective trial demonstrated that daily treatment with a relatively low calcitriol dose (0.25 mcg) effectively negativized IAA and GAD antibodies within 0.6 (0.4–2.1) years in all 12 seropositive children (5 IAA, 6 GAD, 1 IAA+ GAD) aged 6.6 (1.5–13) years at inclusion [39]. They were deemed high risk and tested for T1Ab because of the already diagnosed association of celiac disease and autoimmune thyroiditis (four girls), the onset of autoimmune thyroiditis at a very young age (<3 years of age in two girls and one boy; <5 years of age in one boy), the occurrence of T1D in their siblings (two boys), and impaired glucose tolerance (IGT; one boy, one girl). Calcitriol was discontinued 1 year after negativation of T1Ab while continuing optimal cholecalciferol repletion aiming at serum concentrations of 50–80 ng/mL for up to 4.8 years at the time of publication. Ten years later, none of the patients has developed T1D or other T1Ab, and one with positive IAA relapsed twice but negativized again on 0.5 mcg daily oral calcitriol. Meanwhile, in a 10.5-year-old boy referred for fasting hyperglycemia and found with prediabetes and impaired glucose tolerance (IGT) with all four (ZnT8 unavailable at the time) positive T1Ab and predisposing HLA DR 3/4, interception of clinical disease was accomplished with tolerated high doses of oral calcitriol (up to 6 mcg/day) and mega doses of paricalcitol (up to 72 mcg/day) for over 3 years [40]. This case implies that once pre-T1D is there, vitamin D, calcitriol and its analog, paricalcitol, may indeed delay progression to overt T1D, pointing to the need of research on therapies that could revitalize β-cells, in combination with immune intervention strategies [1], possibly with powerful vitamin D agonists selective for the immune system, even aiming to reverse the disease. It also highlights the importance of early detection of seroconversion towards T1D, as in that stage, immunomodulation with vitamin D, calcitriol and its analogs may prove far more promising and successful.

### 1.5. Aim of the Study

To further investigate whether secondary prevention of Type 1 Diabetes with the active hormone calcitriol and its less calcemic analog paricalcitol is feasible, we initiated a prospective interventional non-randomized clinical trial, the PRECAL study (ISRCTN17354692).

## 2. Materials and Methods

Between 2010 and 2022, 50 children (26 boys, 24 girls) aged 0.65–16.37 years identified to be at high risk for T1D were included in the PRECAL (PREvention of Diabetes Type 1 with oral CALcitriol and analogs) study ISRCTN17354692, implemented by the Pediatric Endocrine and Diabetes Clinic of Athens Medical Center (certified tertiary care medical provider, “https://temos-accreditation.com/AccreditedPartners/Details.aspx?id=6447 (accessed on 15 March 2023)”), after obtention of parental written informed consent.

A STARD (STAndards for the reporting of Diagnostic accuracy studies) shows the disposition of subjects participating in the PRECAL study (Figure 1). A total of 2385 children were screened either because of T1D in a first-degree relative (FDR, sibling or parent), or the development of Hashimoto’s thyroiditis, or the association of Hashimoto’s thyroiditis and celiac disease, or altered glucose metabolism (fasting plasma glucose repeatedly >100 mg/dL, and/or borderline or abnormal HbA1c > 5.7%) and/or formal impaired glucose tolerance at the oral glucose tolerance test (2 h sample > 140 mg/dL, IGT), or even formal laboratory diagnosis of preclinical diabetes (verified plasma fasting glucose > 125 mg/dL, or >200 in the 2 h sample at the OGTT, pre-T1D), or clinically presenting cases with T1D, but who did not yet require insulin replacement at the time of diagnosis. Of 72 high-risk children (3%) that met the inclusion criteria, 50 agreed to enter the study. Those included 9 with T1D who did not require insulin at inclusion, 4 with pre-T1D, 9 with various IGT, 14 FDR (5 with a T1D mother, 4 with a T1D father, and 5 with a T1D sibling), 5 with the association of Hashimoto’s thyroiditis and celiac disease, 6 with Hashimoto’s thyroiditis (all aged <3 years at diagnosis), and 3 with celiac disease. Of the 50 included children, 44 were positive for at least one T1Ab (IAA, IA2, GAD, ICA, ZnT8 if available, T1Ab+), and six were with predisposing HLA A, DQ, DR haplotypes and genotypes (HLA+), tested if they were negative for all available T1Ab. We were able to measure ZnT8 only after 2020 due to previous unavailability of a commercially authorized kit. Regarding HLA risk, any high-risk child as defined above that was negative to the available T1Ab and had a predisposing HLA epitope(s) regardless of the specific risk for this genotype was included in the study. Inclusion criteria proved T1D susceptibility regardless of the glucose metabolism status except for children already diagnosed with T1D, already under insulin replacement therapy, who were excluded from the study. Of the children included, 9 had varied impaired glucose tolerance (IGT, all T1Ab+), 4 were with pre-T1D (3 T1Ab+ and 1 HLA+), and 9 had new-onset T1D (T1Ab+), but they were not on insulin replacement yet. Serum T1Ab levels and fasting plasma glucose, HbA1c, c-peptide, serum Ca (tolerated ≤ 11.5 mg/dL), P, ALP, U, Cr, Ca/Cr 2 h urine morning sample, 25(OH)D, 1–25(OH)2D with renal ultrasound in cases with hypercalcemia/hypercalciuria (tolerated ≤ 50%, or according to normal for age) were determined before and q3–6 months after initiation of calcitriol (0.05 mcg/Kg/day) 0.25–0.5 × 1–3 times/day or its synthetic analog, paricalcitol (thrice the calcitriol dose), 1–4 mcg × 1–3 times/day p.o. along with cholecalciferol repletion using the doses that the Endocrine Society practice committee characterizes as not requiring medical supervision to prevent vitamin D deficiency, practically identical to the former IOM’s (Institute of Medicine) upper tolerable limits: up to 1000 IU/d for infants aged <6 months, 1500 for age 6 months–1 year, 2500 for 1–3 years, 3000 for children 4–8 years and 4000 for children > 8 years to maintain circulating concentrations of 25(OH)D at the optimal range of 40–60 ng/mL [41].

All patients who started on calcitriol or paricalcitol were advised to drink enough water during the day and parents were notified for possible adverse events related to hypercalcemia–hypercalciuria. Within 1–4 weeks after treatment initiation or dose alteration, every patient underwent a fasting morning serum Ca measurement along with a 2 h morning urine sample for Ca/Cr ratio to ensure safety and allow dose titration. In all asymptomatic patients, serum Ca ≤ 11.5 mg/dL and a 2 h morning urine Ca/Cr < 50% or that normal for the age (normal values: <6 months: <0.8, 6–12 months < 0.6, 1–2 years < 0.81, 1–3 years < 0.53, 4–5 years < 0.39, 6–7 years < 0.28, and >7 years < 0.21 [42]) were overall tolerated, provided a normal renal ultrasound by an experienced pediatric radiologist and plasma measurements of urea and creatinine remained within normal range for age. Otherwise, treatment was discontinued or titrated appropriately to meet those requirements. Once negativation of T1Ab was achieved and maintained for 12 months, calcitriol or paricalcitol was discontinued while on daily cholecalciferol supplementation aiming at 25(OH)D concentrations of 50–80 ng/mL under a 3/6-month follow-up for timely detection of any relapses. Children in the HLA predisposing group with normal calcium metabolism received calcitriol 0.5 mcg × 1 time daily p.o., to prevent seroconversion. Therapy cessation in this group was scheduled to occur after a minimum of 3 years, provided a normal glucose and calcium metabolism and T1Ab remained negative. In children with dysglycemia with an abnormal fasting plasma glucose checked on two different occasions (>100 mg/dL) and/or borderline HbA1c (>5.7%) or IGT in a 2 h oral glucose tolerance test (OGTT), calcitriol or paricalcitol dose was titrated, aiming to achieve complete normalization of glucose metabolism.

In children positive for T1Ab, measurements of antibody titers were verified within 1 month, prior to the inclusion in the study and treatment initiation. Treatment selection, dose and duration were individualized and titrated depending on age and safety, the presence or absence of T1Ab and their titers, the presence or absence of the already altered glucose metabolism and the time needed for achieving T1Ab negativation, as well as cost and availability. Calcitriol and paricalcitol were prescribed electronically after an authorized off-label use by the National Organization for Medicines (EOF) with 75% coverage of the cost by the patients’ social security. However, while calcitriol is particularly cheap, recurrent shortages exist in supply, imported only by The Institute of Pharmaceutical Research and Technology (IFET), whereas paricalcitol (Zemplar, Abbvie Pharmaceuticals, Chicago, IL, USA) is particularly costly (i.e., 28 caps of 1 mcg have a retail price of EUR 60, with 25% patient involvement) through the e-EFKA electronic prescription system.

All measurements were performed as previously described [39]. Commercially available ELISA kits were used: EIA-1594, EIA-1910 and EIA-1593 for ICA, anti-GAD65 and IAA, respectively (DRG Diagnostics, Marburg, Germany), and Medizym 3803 for anti-IA2 and 3791 for ZnT8 (Medipan, Berlin, Germany). All enrolled children had a 3–6-month follow-up with detailed physical examination and laboratory investigations. 

Additionally, apart from T1Ab, autoantibody titers against thyroglobulin and thyroid peroxidase as well as anti-transglutaminase IgA and IgG antibodies were also evaluated every 6 months. The main outcome was the reversion of seroconversion, that is, the negativation of the T1Ab(s), or non-progression to an antibody positive status for the HLA+ children.

Statistical analyses were performed in XLSTAT PREMIUM version 2023.1.4 (copyright Addinsoft). Measurements are given as median and range in parenthesis due to the small number of children in the various subgroups analyzed. Direct analysis between groups was performed with the use of the two-sample one-sided t-test. Significance was set as a *p* value < 0.05. A sample size calculation was performed. We used the XLSTAT.ai (artificial intelligence) feature to perform multivariate multiple regression analysis for the time to negativation related to the age at inclusion, sex (male, female), 25(OH)D concentrations at inclusion, the magnitude of positivity, i.e., the titter of T1Ab, the × times from positivity limit of the positive T1Ab(s), the type of treatment (calcitriol or paricalcitol) and the δ change in the magnitude (relative to 1) of the T1Ab concentrations between inclusion and at the time of negativation.

Given that 3% of the screened population for inclusion criteria were found at risk for developing T1D, a sample size calculation was performed. A total of 44 subjects are theoretically needed with an error margin of 5% to reach statistical significance in the intervention. 

## 3. Results

### 3.1. Aggregated Data (Table 1)

Available data on 42 children (7 dropouts, 1 with follow-up <3 m) are presented. In all 26 T1Ab-positive children, with a single T1Ab (24/26, 15 with positive IAA, 3 with positive anti-IA2, 4 with positive ICA, 2 with positive anti-GAD) or with multiple Τ1Abs (2/24, 1 with positive IAA/anti-GAD, 1 with positive ICA/anti-GAD) who did not have pre-T1D or new-onset T1D at the time of inclusion, reversion of seroconversion was achieved within 0.57 years (0.32–1.3) with a total follow-up of 3.06 years (0.5–10). Five children with predisposing HLA but without pre-T1D or new-onset T1D at inclusion did not develop T1D during their follow-up for 3 years (1–4). From four children with pre-T1D, one (#6) with positive T1Ab achieved reversion of seroconversion (follow-up 1 year), one (#50) with predisposing HLA did not develop T1D (follow-up 3.3 years) and two with T1Ab developed T1D within 6 months (#5) and 3 years (#40). From nine children with new-onset T1D (one dropout), three progressed immediately to overt T1D (#1–3), and five (#35–39) had complete remission for 1 year (1 month–2 years). Five children with positive T1Ab who had previously achieved reversion of seroconversion had a relapse after discontinuation of therapy but successfully negativized T1Ab again after resuming therapy. Anti-TPO Abs in three children and anti-TPO/anti-TG in one child, all aged <3 years, were negativized. Anti-TTG-IgA in two children (one child with positive IAA but also with a predisposing HLA aged 2.8 years) and in one positive for IA2/anti-GAD, who progressed to T1D within 1 year, aged 5 years, were also negativized.

**Table 1 children-10-00862-t001:** Patient characteristics and follow-up in the PRECAL study.

#	Gender	Screening Criteria	Age Years	25(OH)D ng/ml	T1Ab(s)	Titer	HLA (PRE/PRO) Predisposing Protective	Treatment Dose (mcg)	Evolution: Insulin/ Years to Negativation/ Drops Out	Titer after Treatment	Follow-Up (Years) after Negativation	Evolution, Relapses, Adverse Events, Thyroid—Celiac Abs Negativation
Single T1Ab-positive								Paricalcitol (P) Calcitriol (C)				
1	Μ	T1D	3.80	43.0	IAA	1.8 < 1.1	PRE A24	C 0.25 × 1	Insulin immediately			
2	M	T1D	2.49	40.0	IAA	1.4 < 1.1	N/A	P 2 × 3	Insulin immediately			
3	F	T1D	9.28	13.1	IA2	394 < 10	N/A	C 0.5 × 1	Insulin immediately			
4	M	T1D	7.80	29.0	IAA	250 < 10	N/A	P 4 × 3	drop out			
5	M	Pre-T1D Hashimoto’s	8.45	24.3	GAD	34.2 < 10	N/A	P 2 × 3	Insulin in 6 months			
6	M	Pre-T1D	11.91	33.1	IAA	1.3 < 1.1	N/A	C 0.5 × 1	1.92	1.0 < 1.1	1.52	
7	F	IGT	10.59	20.0	IAA	10 < 10	N/A	C 0.5 × 1	0.51	7.5 < 10	2.55	Normalized glucose metabolism
8	F	IGT	13.48	18.3	IAA	1.4 < 1.1	PRE A24	C 0.5 × 2	0.32	1.0 < 1.1	2.00	Normalized glucose metabolism
9	M	IGT	14.97	22.1	ICA	1.3 < 1.1	N/A	C 0.5 × 3	0.90	0.9 < 1.1	4.02	Normalized glucose metabolism
10	Μ	IGT	12.25	27.0	ICA	1.1 < 1.1	N/A	C 0.5 × 1	0.48	0.4 < 1.1	1.80	Normalized glucose metabolism
11	M	IGT	4.55	18.7	IAA	2.7 < 1.1	PRE A02	P 6 × 3	0.95	1.0 < 1.1	2.35	Relapse, hypercalciuria
12	M	IGT	8.94	21.3	ΙAA	1.3 < 1.1	N/A	P 2 × 1	0.30	0.9 < 1.1	2.00	Normalized glucose metabolism
13	F	IGT	13.45	15.6	IAA	2.2 < 1.1	N/A	C 0.5 × 1	drop out			
14	F	IGT	4.92	44.8	GAD	47 < 10	N/A	P 1 × 3	0.30	1.2 < 10	0.30	Normalized glucose metabolism
15	M	IGT	6.37	22.1	ICA	1.26 < 1.1	N/A	P 2 × 3	drop out			
16	M	Mother T1D	4.10	18.3	IAA	1.52 < 1.1	PRE DR4	C 0.5 × 2	0.57	0.34 < 1.1	0.60	
17	F	Sibling T1D	11.35	28.3	IA2	16 < 10	N/A	C 0.25 × 3	0.53	4.50 < 10	2.70	
18	F	Sibling T1D	1.31	22.0	IAA	1.39 < 1.1	N/A	P 4 × 3	1.30	0.24 < 1.1	3.24	Relapse, hypercalciuria, hypercalcemia
19	F	Father T1D	4.91	20.0	IA2	37.7 < 10	PRE DR4	P 2 × 3	0.50	2.0 < 10	2.50	
20	M	Father T1D	8.38	38.0	ΙAA	1.8 < 1.1	PRE A02	P 2 × 3	Follow-up <3 m			
21	Μ	Sibling T1D	4.09	18.0	ΙAA	2.9 < 1.1	N/A	C 0.5 × 1	drop out			
22	F	Sibling T1D	10.67	33.8	IAA	1.8 < 1.1	PRE A02/DR4	C 0.5 × 1	1.9	1.0 < 1.1	9.24	Relapse, hypercalciuria
23	F	Hashimoto’s Celiac disease	9.28	27.0	IAA	1.3 < 1.1	N/A	C 0.25 × 1	0.9	0.2 < 1.1	5.20	
24	M	Hashimoto’s Celiac disease	13.17	26.6	GAD	32 < 10	PRE A02	P 2 × 3	0.5	4 < 10	4.13	
25	F	Hashimoto’s Celiac disease	10.04	26.8	ICA	2.4 < 1.1	PRO DQA10102 PRE DQA10501	P 2 × 3	1.0	0.25 < 1.1	1.68	
26	F	Hashimoto’s Celiac disease	12.54	23.8	IAA	1.1 < 1.1	N/A	C 0.5 × 1	0.46	0.5 < 1.1	5.00	
27	F	Celiac disease	4.13	27.8	IAA	1.8 < 1.1	N/A	P 2 × 3	1.25	0.7 < 1.1	2.38	Hypercalciuria, hypercalcemia
28	M	Celiac disease	13.03	17.4	ΙAA	1.1 < 1.1	N/A	C 0.5 × 1	0.60	0.9 < 1.1	5.86	
29	F	Celiac disease	2.84	12.5	ΙAA	1.4 < 1.1	PRE DQA10105	C 0.5 × 1	1.60	0.9 < 1.1	2.70	Relapse, hypercalciuria, negativized celiac disease Abs
30	F	Hashimoto’s	3.28	19.5	ICA	1.5 < 1.1	PRE A02	C 0.5 × 1	0.37	0.1 < 1.1	0.37	
31	Μ	Hashimoto’s	2.18	19.0	IA2	16.2 < 10	N/A	C 0.5 × 1	1.20	8.5 < 10	6.80	Negativized thyroid Abs
32	F	Hashimoto’s	0.65	22.0	IAA	1.5 < 1.1	N/A	P 1 × 2	0.30	0.9 < 1.1	2.60	Hypercalciuria, hypercalcemia, negativized thyroid Abs
33	M	Hashimoto’s	1.40	30.4	IAA	8.3 < 1.1	N/A	P 1 × 3	0.50	0.52 < 1.1	0.90	Hypercalcemia, negativized thyroid Abs
34	F	Hashimoto’s	2.47	17.0	ΙAA	1.2 < 1.1	N/A	C 0.25 × 1	drop out			
Multiple T1Ab positive									N/A			
35	F	T1D	4.37	18.7	IAA GAD	1.2 < 1.1 54 < 10	N/A	P 2 × 3	Insulin in 0.80 years			
36	Μ	T1D	16.37	18.3	ΙA2 GAD	37.5 < 10 188 < 10	PRE DR4	P 12 × 3	Insulin in 0.85 years			
37	F	Τ1D	6.65	22.7	IAA ICA GAD	1.3 < 1.1 1.2 < 1.1 79 < 10	PRE DRB10404	C 0.25 × 1	Insulin in 2 years			
38	F	T1D	5.09	19.6	IA2 GAD	400 < 10 125.2 < 10	N/A	P 8 × 3	Insulin in 0.95 years			Negativized Celiac disease Abs
39	F	T1D	6.10	25.7	ΙCA GAD	4.75 < 1.1 280 < 10	N/A	P 2 × 3	Insulin in 1 week			
40	M	Pre-T1D	10.53	22.0	IAA GAD	1.3 < 1.1 93 < 10	N/A	C 0.5 × 3	Insulin in 3 years			
41	M	Hashimoto’s	1.85	15.8	IAA GAD	1.3 < 1.1 12 < 10	N/A	C 0.25 × 1	0.41	0.5 < 1.1 6.0 < 10	8.05	Negativized thyroid Abs
42	Μ	Sibling T1D	1.90	21.0	ICA GAD	1.2 < 1.1 12 < 10	N/A	P 1 × 3	0.58	0.44 < 1.1 4.0 < 10	1.00	Relapse
43	M	Mother T1D	11.32	20.7	IAA ICA	1.2 < 1.1 1.2 < 1.1	N/A	C 0.5 × 1	Drop out			
44	Μ	Mother T1D	4.90	25.5	ΙAA GAD	1.2 < 1.1 1.2 < 1	N/A	P 2 × 1	Drop out			
HLA Predisposing												
45	F	Mother T1D	8.30	13.0	-		PRE A24	C 0.5 × 1			4.5	
46	F	Mother T1D	10.29	15.0	-		PRE A24	C 0.5 × 1			4.5	
47	M	Sibling T1D	2.49	30.8	-		PRE A24	C 0.5 × 1			1.0	
48	F	Father T1D	11.78	19.5	-		PRE A24, DR3	C 0.5 × 3			1.0	
49	M	Hashimoto’s Celiac disease	12.25	16.0	-		PRE A02	C 0.5 × 1			1.0	
50	M	Pre-T1D	3.88	29.9	-		PRE A02	C 0.25 × 2			3.5	Hypercalciuria

N/A: non-available.

### 3.2. Single T1Ab-Positive Children

From 28 single T1Ab-positive children (34 initially included, 5 dropouts, 1 with follow-up <3 months) aged 9.28 years (0.65–14.97), 3 (2 boys and 1 girl) had new-onset T1D not on insulin yet, 2 boys had pre-T1D, and 9 had IGT. Four children from these groups (with new-onset T1D and pre-T1D) progressed to overt T1D. Three of them, two with positive IAA (#1–2), and one with positive anti-IA2 (#3), who had presented with new-onset T1D which progressed immediately after diagnosis to overt disease, and one boy (#5) with positive anti-GAD with pre-T1D at inclusion progressed to overt T1D within 6 months. In the remaining 24 single T1Ab-positive children (15 with positive IAA, 3 with positive anti-IA2, 4 with positive ICA, and 2 with positive anti-GAD), reversion of seroconversion was achieved by all of them within 0.57 years (0.3–1.92) after initiation of calcitriol (*n* = 14) or paricalcitol (*n* = 10) treatment. Seven children (five boys and two girls) have already completed calcitriol (*n* = 4, #23, #26, #28, #31) or paricalcitol (*n* = 3, #12, #24, #33) treatment after reversion of seroconversion of T1Ab being achieved in 0.5 years (0.3–1.2 years), with a median duration of treatment of 2.5 years (0.5–3.5). There have been no relapses for 5 years (0.9–6.8) during follow-up. Six children, four girls and two boys, five on calcitriol (#`8, #11, #16–17, #30) and one on paricalcitol (#27) in whom reversion of seroconversion was achieved within a period of 0.5 years (0.32–1.25), with no relapses reported during or after follow-up, discontinued their follow-up after 1.9 years (0.37–2.7). The remaining 11 children (8 girls and 3 boys) are still on calcitriol (*n* = 5, #6–7, #9, #22, #29) or paricalcitol (*n* = 6, #11, #14, #18–19, #25, #32) treatment. Reversion of seroconversion has already been achieved in all of them at a median of 0.97 years (0.3–1.92) with a post-negativation follow-up of 2.65 years (1.52–9.24 years).

#### Relapses in Single T1Ab-Positive Children Who Achieved Reversion of Seroconversion

There were four relapses in this group. The first occurred in a girl positive for IAA, sister of a T1D girl (#22). She was started on calcitriol 0.5 mcg/day at the age of 10.5 years and reversion of seroconversion occurred within 2 years of treatment. During follow-up and while still on calcitriol, she relapsed after 3 years of treatment. Calcitriol was then increased to 0.5 mcg × 3 times/day leading to successful negativation of the IAA for 2 more years. Then, an attempt to decrease calcitriol dose to 0.5 mcg/day resulted in a second relapse. The antibody titer negativized again after 6 months on paricalcitol 2 mcg × 3 times/day. The dose was then titrated to 2 mcg × 2 times/day. For the last 2.3 years, she has remained negative for T1Ab, having completely normal calcium metabolism under cholecalciferol 4000 IU/day and paricalcitol 2 mcg × 2 times/day. The second relapse ensued in a girl who was detected positive for IAA at the age of 2.84 years due to positive anti-TTG-IGA Abs (#29). She was initially started on calcitriol 0.5 mcg daily, increased to 0.5 mcg twice daily, because of an increase in the IAA titer after 1 year. Negativation of IAA was achieved after 1.6 years of calcitriol therapy and the dose was gradually decreased to 0.25 mcg daily. While for 2 years on this low dose of calcitriol, a new relapse occurred, with IAA rising again to 65.2 (<20), and she was then switched to paricalcitol 1 mcg thrice daily, still on treatment. The third case of relapse was an IAA-positive girl aged 1.31 years at inclusion (#18), sibling of a T1D child, who negativized after being 1.3 years on paricalcitol 1 mcg × 3 times/day. She remained negative with no treatment for 1.8 years and then a relapse occurred, and she was started on calcitriol 0.5 mcg/day. She then again achieved reversion of seroconversion within 0.5 years. The fourth case of relapse was an IAA-positive boy (#11) who was first screened due to IGT (increased HbA1c), and he was started on paricalcitol 2 mcg × 3 times/day, gradually increased to 6 mcg × 3 times/day to achieve reversion of seroconversion, which occurred after 0.95 years. Then, the dose was gradually decreased and stopped 2 years later due to minor signs of nephrocalcinosis in the renal ultrasound. The antibodies remained negative for 1.8 year, when relapse occurred again (IAA 124, <20). We then restarted paricalcitol 2 mcg × 3 times/day, currently waiting for the first results after 3 months.

### 3.3. Multiple T1Ab Positive Children

Eight multiple T1Ab-positive children (ten at inclusion, two dropouts) aged 5.59 years (1.85–15.37) received either calcitriol (*n* = 3) or paricalcitol (*n* = 5). Five of them (three girls #35, #38–39 and one boy (#36) with two positive T1Ab and one girl #37 with three positive T1Ab) already had T1D at inclusion. Insulin replacement was started within 0.9 years (0.8–2), apart from a girl (#39) in whom COVID infection one week after paricalcitol induction led to high blood sugar levels, requiring immediate initiation of insulin replacement therapy. One boy with pre-T1D (#40) and two with positive T1Ab finally progressed, after 3 years, to overt T1D. The remaining two children, each with two positive T1Ab aged 1.85 (#41) and 1.9 years (#42), achieved reversion of seroconversion within 0.41 and 0.53 years, on calcitriol and on paricalcitol, respectively. The first is still on calcitriol 8.05 years after inclusion. He was started on a low dose of 0.25 mcg/day, increased to 0.5 mcg/day 2.53 years later, after a relapse in the IAA titer. The second child stopped treatment after achievement of seroconversion (parental decision) and he remains negative 1 year from inclusion while on cholecalciferol repletion maintaining 25(OH)D concentrations 50–80 ng/mL and a completely normal Ca metabolism. 

### 3.4. Children with Predisposing HLA

From the six children (three girls and three boys) aged 9.29 years (2.49–12.25) with predisposing HLA, three (one boy aged 2.49 years #47, one boy aged 12.25 years #49 and one girl aged 11.78 years #48 at inclusion) discontinued their follow-up after 1 year. During this time, they all remained negative for T1Ab. Two sisters (mother with T1D) aged 8.36 (#45) and 10.29 years (#46) at inclusion are still on calcitriol 0.5 mcg/day and remain negative for 4.5 years now. In a boy aged 3.88 years (#50) at inclusion who had pre-T1D with abnormal OGTT (formal IGT), while still being negative 3.5 years after inclusion, switching to paricalcitol (1 mcg × 3 times /day) from calcitriol 0.5 mcg/day normalized glucose tolerance within 9 months.

### 3.5. Observed Effects on Thyroid and Celiac Disease Related Abs

Negativation of autoantibodies against thyroglobulin and/or thyroid peroxidase was observed in four children with Hashimoto’s thyroiditis (one girl and three boys) of a total of eleven children diagnosed with Hashimoto’s. One girl aged 0.65 years (#32) positive for non-maternal anti-TPO Abs (17, <14), one boy aged 2.18 years (#31) positive for both anti-TPO (26.5, >14) and anti-TG (28.8, <8), and one boy aged 1.38 years (#33) positive for anti-TPO (12.3, <8) were positive to one T1Ab, and the other boy aged 1.85 years (#41) positive for anti-TPO (44.2, <14), had multiple T1Ab. All four negativized their thyroid Abs titers by 3 months after initiation of paricalcitol 1 mcg × 2 times/day in the girl, 5 months after paricalcitol induction at 1 mcg × 3 times/day in the 2.18-year-old boy, and by 6 months after initiation of calcitriol 0.5 mcg × 1/day in the remaining two boys.

Anti-TTG-IgA Ab titer evolution could not be assessed in children that initiated a gluten-free diet as soon as the diagnosis of celiac disease had been established. However, two girls, aged 2.8 years (#29) with positive IAA, and 5 years (#38) with positive anti-IA2/anti-GAD who progressed to overt T1D within a year had mild elevation of anti-TTG-IgA Abs (15.8, and 27.5, <10, respectively), did not start on a gluten-free diet, and achieved reversion of seroconversion by 3 months and 9 months on calcitriol and paricalcitol treatment, and remain negative for 4.3 and 2 years, respectively.

### 3.6. Vitamin D Status

The majority—but not all children—were vitamin D-deficient (25(OHD) < 20 ng/mL) or insufficient (25(OHD)D < 30 ng/mL) at inclusion. In children with only one positive T1Ab, median 25(OH)D concentration was 22.1 (13.1–44.8) ng/mL, in those with multiple positive T1Ab it was 20.8 (15.1–25.7) ng/mL, and in those with a predisposing HLA it was 17.75 (13–30.8) ng/mL, which is not statistically different, possibly due to the small number of patients. In the single T1Ab group, from a total of 34 children, 7 were vitamin D-sufficient, with 3 having concentrations of >40 ng/mL and 4 > 30 ng/mL at inclusion. All children in the multiple T1Ab group were vitamin D-deficient, though, and only one child was vitamin D-sufficient in the HLA predisposing group. 

### 3.7. Adverse Events: Hypercalcemia, Hypercalciuria and Nephrocalcinosis

In eight children (five girls and two boys with a single T1Ab, and one boy with predisposing HLA), hypercalciuria and/or hypercalcemia with otherwise normal renal function was observed during treatment. One girl from the single T1Ab group aged 10.67 years (#22) developed mild transient hypercalciuria (Ca/Cr 0.55, <0.21) after being on calcitriol 0.5 mcg × 1/day for 1.48 years, which resolved spontaneously with no dose titration or cessation. Renal ultrasound was normal. She is still on calcitriol therapy with completely normal calcium metabolism. In three girls from the single T1Ab group aged 1.87 (#18), 4.73 (#27) and 0.98 years (#32), hypercalciuria Ca/Cr 0.33, 1.37, 0.84 and hypercalcemia 12, 13.4, 11.5 mg/dL were observed by 0.5, 0.6 and 0.33 years of paricalcitol 4 mcg × 3 times/day, 2 mcg × 3 times/day and 2 mcg × 2 times/day, respectively. In the first girl, hypercalcemia and hypercalciuria resolved after dose titration at 2 mcg × 3 times/day and remained normal until therapy cessation 1.2 years later. In the second girl, hypercalcemia/hypercalciuria improved after paricalcitol dose titration at 2 mcg × 2 times /day (calcium 10.7, Ca/Cr 0.74), but a further dose reduction to 2 mcg × 1 time/day was needed to achieve complete calcium metabolism normalization. The third girl normalized calcium metabolism after paricalcitol dose titration at 2 mcg × 1 time/day. All three had a normal renal ultrasound. Another girl aged 3.45 years (#29) presented mild hypercalciuria (Ca/Cr 0.36) without hypercalcemia at 0.66 years of calcitriol treatment (0.5 mcg × 2 times/day), but as the IAA Abs, positive at inclusion, had not negativized yet, it was decided to tolerate it after performing a renal ultrasound, which was normal. One year later, reversion of seroconversion was achieved, and the calcitriol dose was reduced to 0.5 mcg/day. At that time, serum calcium was 9.7 mg/dL and urine Ca/Cr was 0.5. Calcitriol was further reduced at 0.25 mcg/day after 7 months and maintained for 2 years when a relapse occurred, which led us to switch on paricalcitol 1 mcg × 3 times/day. One boy aged 4.55 years (#11) at inclusion from the single T1Ab group presented mild hypercalciuria (Ca/Cr 0.37) without hypercalcemia after 1.12 years on paricalcitol when we gradually titrated the dose from 2 mcg × 3 times/day to 6 mcg × 3 times/day to achieve reversion of seroconversion one year later. Despite the mild hypercalciuria, nephrocalcinosis grade II was detected, and we stopped paricalcitol as soon as reversion of seroconversion was achieved. The T1Ab remained negative for 1.8 years, and nephrocalcinosis disappeared when a relapse occurred with high IAA titers 124, <20. He was restarted on paricalcitol 2 mcg × 3 times/day, currently waiting for the first results after a 3-month period. The last boy from the single T1Ab-positive group aged 1.4 years (#33) at inclusion with a particularly high IAA titer (150, <20) developed hypercalcemia 16 mg/dL after 0.5 yrs on paricalcitol 0.1 mcg × 3 times/day, but as he had already negativized IAA, treatment was stopped, and calcium metabolism normalized, and the renal ultrasound was normal. He is still IAA negative after 4 months, receiving only cholecalciferol repletion to maintain 25(OH)D concentrations of 50–80 ng/mL. One more boy from the HLA predisposing group (#50) presented hypercalciuria (Ca/Cr 0.67) 9 months after being on calcitriol 0.25 mcg × 2 times/day. The renal ultrasound was normal and calcium metabolism normalized completely after switching to paricalcitol 2 mcg × 1 time/day.

### 3.8. Statistical Analysis

Given the population of 2696 children checked (*n* = 2696) after excluding clinical T1D cases (*n* = 2385), 72 children were found eligible constituting a population proportion of 3% (Figure 1), and 44 children are needed to achieve statistical significance according to the study design. Analyzing together all healthy children that achieved negativation of T1Ab at inclusion, 24 with one positive T1Ab (patients #6–12, 14–19, 22–33) and 2 with two positive T1Ab (patients #41–42), the best model, according to the XLSTAT.ai regression analysis, was the Random Forest regression with a mean squared error (MSE) computed as 0.224 on the validation sample, in the lowest of the ideal range of 0.2–0.5. One of the most important features of the Random Forest Algorithm is that it can handle the data set containing continuous variables, as in the case of regression, and categorical variables, as in the case of classification. Thus, it performs better for classification and regression tasks. Regarding the δ change in T1Ab magnitude (concentrations relative to 1) between inclusion and at the time of negativation (*p* < 0.001) and the time needed to negativation, the variable importance was for age at inclusion −1.944, 25(OH)D concentrations at inclusion −4.036, magnitude of the T1Ab at inclusion +0.752, inclusion criteria −0.150, and type of T1Ab −0.961, meaning that practically only the titers of T1Ab at inclusion seem to have an impact on the time to negativation. However, using a Cox proportional hazard model with the time to negativation of T1Ab as the outcome variable, as well as age, gender, initial concentrations of 25(OH)D, concentrations (as magnitude) and type of T1Ab, and kind of treatment (calcitriol or paricalcitol) as the covariates, none was significantly correlated with the time to negativation of T1Ab. The proportionality assumption was checked with the Schoenfeld and scaled Schoenfeld residuals. The Schoenfeld tests showed that there was no violation of the proportional assumption in our model (all *p* > 0.05). 

## 4. Discussion

In the preliminary results (2010–2022) of the ongoing PRECAL study, children positive for T1Ab were treated with oral calcitriol or paricalcitol. We acknowledge that this is not a randomized double-blind controlled trial but a prospective interventional study (2010–2030) aiming not only to verify and extend the results of a previously published pilot trial [39], but also to vigorously protect children at risk included. The results presented, however, indicate that treatment with calcitriol and its analog paricalcitol may be effective in preventing T1D, resulting in reversion of seroconversion with negativation of even multiple T1Ab in all healthy high-risk children included, at least if started soon enough after seroconversion.

The limitations are numerous, the most important being the relatively small number of patients in the different subgroups of patients (single positive T1Ab, multiple positive T1Ab, HLA+) who were also at various ages and clinical stages, and the absence of a placebo-treated control group, which would encounter, apart from feasibility, ethical issues as well. 

Ideally, early prediction of childhood T1D may prevent the incidence of ketoacidosis at diagnosis, but also provide a window of opportunity for disease prevention [43], as the recent promising results of teplizumab delaying progression to clinical T1D in high-risk subjects demonstrated [44]. In prospective studies on islet autoimmunity, the appearance of GAD peaked as early as 33 months, later than that of IAA, which peaked already at 18 months [9]. Genetic factors alone cannot explain the rapid increase in incidence observed worldwide, and one of the highest postulated mechanisms is hours of sunshine, translating to vitamin D concentrations [45]. A marked increasing incidence trend over the past two years could be related to the global impact of the COVID-19 pandemic [46], which may have worsened the population’s vitamin D status [47]. 

One could argue that in our cohort, some children included, especially the ones who were positive to just one T1Ab, may not have developed T1D at a later stage. Even if this stands, it does not reduce the significance of the successful reversion of seroconversion with achieving negativation of the T1Ab titer and the repeated negativation of relapsed cases with re-initiation of treatment in all healthy children. To further evaluate this aspect, we compared the antibody titters of the children in the present study with those in children diagnosed with T1D during the same period in our department. Of the 71 children (29 girls and 42 boys; 0.6–17 years of age) with newly diagnosed T1D, 16 were not positive to any T1Ab (ZnT8 unavailable), 29 were positive to only one T1Ab, 18 were positive to two T1Ab, and 7 were positive to three T1Ab. In detail, 28 children were positive for GAD, 16 for IAA, 15 for ICA, and 31 for IA2. The median titer of IAA was 1.6, for IA2 it was 183.5, for GAD it was 10,6 and for ICA it was 1.4 in the T1D group and 1.35, 80.5 and 1.3, respectively, in the seropositive children receiving calcitriol or paricalcitol and did not differ significantly. This means that although T1Ab are related to the risk of T1D, they are not the primary mediators, and their presence or titers do not directly correlate with the degree of β-cell dysfunction [48].

According to the recommendations issued in 2001 by the Immunology of Diabetes Society [43], strategies for full risk evaluation should include determination of at least three of the four best-established markers, namely IAA, ICA, GAD and IA2. The additional major autoantigen against the zinc transporter 8 (ZnT8) was identified only in 2007 and was not widely available in commercially available kits for clinical use until recently [49]. Of course, the high predictive value of T1Ab in first-degree relatives decreases in the general population, where the incidence of T1D is lower, as several large prospective studies have shown. The Karlsburg T1D risk study on schoolchildren [50], aiming to evaluate the predictive diagnostic value of T1Ab in the general population, determined that of a total of 11,840 schoolchildren tested for all four T1Ab, 6.9% were positive for a single T1Ab and only 0.7% were positive to multiple T1Ab. These findings suggest that combined autoantibody screening in the general population clearly identifies those at high risk for T1D. The number of auto-antibodies, rather than the individual antibody, may be most predictive of the imminent risk of progression to overt diabetes [43]. However, T1Ab can both appear and disappear in follow-up samples [51], and a substantial number of patients who are positive for a single T1Ab may not eventually develop T1D. For the initial screening for T1D autoantibodies to be sensitive and efficient for public health translation, it must be performed at ages of 2 and 6 years as the Type 1 Diabetes Intelligence (T1DI) cohort showed [37]. Namely, testing for T1Ab at 2 years and 6 years of age had the highest sensitivity (82%) and positive predictive value (79%) for diabetes by the age of 15 years. That is, 82% of the individuals who develop T1D by the age of 15 years can be identified by this targeted autoantibody screening, and 79% of the participants identified as being T1Ab positive ultimately develop T1D by 15 years of age [52].

In the PRECAL study, most but not all children were vitamin D-insufficient or -deficient. Children with multiple T1Ab had the lowest median vitamin D concentrations, even if not statistically significant possibly due to the small numbers of patients involved compared to the single T1Ab and predisposing HLA groups. Numerous studies demonstrated the association between hypovitaminosis D and T1D [53,54] and islet cell autoimmunity [55], while others suggested that vitamin D supplementation in children reduces the risk of T1D. For example, the Diabetes Autoimmunity Study in the Young (DAISY) reported that the maternal dietary intake of vitamin D was significantly associated with a decreased risk of the appearance of islet autoantibodies in offspring, independent of the human leukocyte antigen (HLA) genotype, family history of T1D, the presence of gestational diabetes mellitus, and ethnicity [56]. Increasing vitamin D intake during pregnancy reduces the incidence of islet autoantibodies in offspring [57], and cod liver oil taken during the first year of life reportedly reduces the risk of childhood onset T1D [58]. A meta-analysis that included observational studies and randomized controlled trials suggested that vitamin D supplementation in infancy may decrease the risk of T1DM in later life [59]. Another meta-analysis of observational studies demonstrated that supplementation of vitamin D in early life reduced risk of developing T1D by 5% if taken in gestation, and significantly by 29% if taken in early infancy [60]. Moreover, a study from Denmark demonstrated that exposure to vitamin D-fortified margarine during the first year of life reduced the risk of T1D in males aged up to 14 years by 2–7.5 times [61]. Together, these findings suggest that by preventing vitamin D deficiency and by using the immunomodulatory effects of vitamin D, primary prevention of T1D might be possible. However, in the PRECAL study, from the seven vitamin D-sufficient children in the single T1Ab group, two with 25(OH)D concentrations of > 40 ng/mL progressed to overt T1D immediately, one with >40 ng/mL had IGT, and the other three vitamin D-sufficient children had undergone seroconversion towards T1D. These data suggest that maintaining vitamin D sufficiency may not be enough to prevent seroconversion towards T1D, opening the discussion for the need of optimal vitamin D concentrations, i.e., 50–80 ng/mL, as even in vitamin D-sufficient children reversion of seroconversion and normalization of glucose metabolism can be achieved through treatment with the active hormone calcitriol and its analog, paricalcitol.

Both pancreatic insulin-producing β-cells and immune system cells express the vitamin D receptor (VDR) and the vitamin D-binding protein (VDBP), polymorphisms in which have been demonstrated to be involved in the pathogenesis of T1D in genetically susceptible individuals [62,63,64,65,66]. Decreased concentrations of VDBP have been reported in both T1D patients and pregnant women whose offspring developed T1D [67,68]. Moreover, many studies showed a positive effect of different forms of vitamin D with significant reduction in insulin needs in supplemented groups with T1D [69,70,71,72,73]. Significant decrease in HbA1c levels occurred after 3 months of treatment with cholecalciferol at different doses in children and adolescents with T1D [74]. Consistently with these findings, eight children in the PRECAL study presented either normalization of or significant improvement in glucose metabolism, treated with calcitriol or paricalcitol, and they had various impaired glucose metabolism at inclusion (IGT): formal impaired glucose tolerance at the OGTT, elevated HbA1c > 5.7% or persistently elevated fasting plasma glucose of >100 mg/dL. 

Apart from T1Ab negativation, some children, all younger than 3 years of age, also negativized anti-TG, anti-TPO and anti-TTG IgA, implying that reversal of seroconversion in younger children possibly provides additional protection from Hashimoto’s and celiac disease. It is known that 1,25(OH)2D3 has potent direct effects regulating autoantibody production [75] while modulating inflammation [76], but without disturbing the antibody response to infections and thus not being immunosuppressive [77]. Moreover, it has been proposed that low pre- and perinatal vitamin D levels imprint on the functional characteristics of various tissues throughout the body, predisposing the affected individual to an increased risk of developing a range of adult-onset disorders, proposing a critical window during which vitamin D may have a persisting impact on adult health outcomes [78].

Taken together, the results of the preceded pilot clinical trial [39] and the PRECAL study, all children who had not developed preclinical disease either with one or multiple T1Ab successfully achieved reversion of seroconversion on calcitriol or paricalcitol and did not progress towards the development of T1D. In the PRECAL study, from four patients with pre-T1D, one negativized T1Ab, one presenting predisposing HLA+ did not progress to T1D (follow-up 3.3 years) and two were positive for T1Ab developed T1D in 6 months and 3 years. Indeed, supplementation with oral calcitriol at the low dose of 0.25 mg/day after the onset of T1D did not induce any sustainable beneficial effect in the IMDIAB XIII trial [79], underlying—combined with our data—the importance of intervening as early and as close to seroconversion as possible in order to induce successful immunomodulation and prevent further progress towards T1D.

The following doses of cholecalciferol repletion were proposed and used in the PRECAL study: up to 1000 IU/d for infants aged <6 months, 1500 IU/d for those aged 6 months–1 year, 2500 IU/d for those aged 1–3 years, 3000 IU/d for children aged 4–8 years and 4000 IU/d for children >8 years. These does are absolutely safe with the rare exception of vitamin D hypersensibility [80]. The same doses that the Endocrine Society practice committee characterizes as not requiring medical supervision to prevent vitamin D deficiency are practically identical to the former IOM’s (Institute of Medicine) upper tolerable limits: up to 1000 IU/d for infants aged <6 months, 1500 IU/d for those aged 6 months–1 year, 2500 IU/d for those aged 1–3 years, 3000 IU/d for children aged 4–8 years and 4000 IU/d for children aged > 8 years, with adults, pregnant/lactating women and adolescents requiring a daily intake of 4000–5000 IU/d (8000–10,000 IU/d if obese) to maintain circulating concentrations of 25(OH)D at the optimal range of 40–60 ng/mL [47], with the largest meta-analysis ever conducted showing that reduction in all-cause mortality related to serum 25(OH)D is maintained with 25(OH)D ≥ 50 ng/mL, demolishing the previously reported U-shaped curve [81].

Serum 25-hydroxyvitamin D [25(OH)D] concentrations higher than 150 ng/mL are the hallmark of vitamin D toxicity due to vitamin D overdosing [82]. Hence, none of our patients exceeded this limit. For vitamin D to cause hypercalcemia, the “free” concentration of 1,25(OH)2D must be inappropriately high, and for this displacement of 1,25(OH)2D plasma 25(OH)D concentrations have to become higher than 240 ng/mL [83]. 1,25(OH)2D—the active form of vitamin D and the ligand for the VDR—acts to increase calcium absorption from the intestine. If normal calcium is unable to be maintained by intestinal calcium absorption, or if 1,25(OH)2D is inappropriately high, then both 1,25(OH)2D and PTH, acting via their receptors, release calcium bone and increase reabsorption of calcium from the distal tubule of the kidney, resulting in hypercalcemia–hypercalciuria and possible nephrocalcinosis [84]. Thus, calcitriol administration requires careful surveillance of calcium metabolism. In the PRECAL study, we used the higher calcitriol doses recommended for hypoparathyroidism treatment for children and adolescents [85], divided into one, two or three daily doses, as the half-life of calcitriol varies from 4 to 15 h [86]. When we intended to use higher doses, we chose or switched to thrice this dose in paricalcitol—again, divided into one, two or three daily doses—a commercially available calcitriol analog, three times less potent in activating the VDR but with ten times less calcemic effect [87], and a half-life of 5–7 h in healthy subjects [88]. Parents were notified of the possible adverse event of hypercalcemia–hypercalciuria. Mild hypercalcemia (10.5–11.9) [89] and hypercalciuria [90]—the latter evaluated in a morning 2 h sample as a calcium/creatinine ratio [42]—were tolerated at least for 6 months while awaiting for a change in T1Ab titters, provided the patients were asymptomatic and with a normal renal ultrasound. 

Concerning calcitriol and paricalcitol safety, LD50 studies were performed on both mice and rats and indicated that the approximate oral lethal dose of calcitriol ranges between 1.35 and 3.9 mg/kg (https://www.rochecanada.com/PMs/Rocaltrol/Rocaltrol_PM_E.pdf, accessed 11 February 2023), and the minimum IV lethal dose for paricalcitol of > 16 mcg/kg for rats and >24 mcg/kg for mice (https://safetydatasheets.pfizer.com/DirectDocumentDownloader/Document?prd=PZ03129~~PDF~~MTR~~HSP~~EN~~PFIZER&DocType=DOC, accessed on 11 February 2023), several orders of magnitude higher than the doses used in the PRECAL study.

While most screening programs aiming to identify individuals at risk for T1D target first-degree relatives of patients with T1D, 90% of those who ultimately develop T1D do not have a family history [91]. Once T1Ab are detected, the rate of progression to T1D depends on genetics, number, type and titter of the T1Ab, but also on metabolic parameters such as BMI and insulin resistance [52], with diabetes type-specific genetic risk scores, however, being consistent predictors of diabetes type across racial/ethnic groups [92]. Thus, HLA genes have a strong impact on the appearance of T1Ab not only in first-degree relatives but in the general population as well [92]. Progression from single to multiple T1Ab happens early after seroconversion, regardless of the timing of the first seroconversion. Consequently, early identification of seroconverted children is crucial, especially before the age of 5 years, as those children present not only a higher rate of seroconversion towards multiple T1Ab but also a faster progression to T1D: about 69% of those who convert from single to multiple T1Ab do so within 2 years of their seroconversion, with 85% reaching the multiple T1Ab status within 18 months, this applying also for children with predisposing HLA alleles [93]. In a large cross-sectional study of thyroid autoimmunity, 6.9% of children at genetic risk for T1D had thyroid autoimmunity, overrepresented in girls and positively related to GAD, ZnT8 variants and IA2 as well as to multiple T1Ab. A considerable percentage of children, however, may develop thyroid autoimmunity already at 18 months of age, before or simultaneously with islet autoimmunity [94]. Children with HLA susceptibility to T1D and celiac disease seroconvert to at least one T1Ab at the median age of 3 years and to at least one celiac disease-associated antibody at 18 months [95], which is also the age at which that IAA have already peaked [9]. With all the compelling evidence discussed above, T1Ab screening in children with a positive family history or who develop thyroid and/or celiac disease autoimmunity especially at an early age is absolutely justified. Furthermore, using integrated adaptive strategies with family history, genotyped risk and T1Ab requires at least 50% fewer surveillance evaluations enhancing T1D prediction as it includes children negative to T1Ab who nonetheless may be at high risk of early disease onset, especially at the vulnerable ages of <2 years [96].

Apart from lessening the danger for severe and life-threatening ketoacidosis, for any of these early detection strategies to become a viable public health option, an effective, safe, costless, and widely available prevention approach is needed. “Healing is a matter of time, but it is sometimes also a matter of opportunity” [97]. Medical practice cannot rely only on plausible theories, but on experience combined with reason. Thus, the Hippocratic logic [98] would probably suggest blind supplementation of pregnant women, newborns, babies, children and adults with the safe cholecalciferol doses discussed above do not require medical supervision according to the Endocrine Society Expert Committee and are considered as the upper tolerable doses by the former IOM. The effect of such a strategy overall in human health is probably inestimable at this time, although Garland et al. provided us with a very good picture of what could be the benefit [81]. The specific effect in T1D incidence, however, has already been practically calculated by Hyppönen et al. [34].

## 5. Conclusions

No agent has yet been shown to be more effective in preventing T1D than vitamin D. Ultimately, primary prevention of islet autoimmunity will likely be the optimal approach for the prevention of T1D, with timely and optimal vitamin D supplementation. At the doses used in the PRECAL study, calcitriol and its less calcemic analog, paricalcitol, were 100% effective in negativizing T1Ab in healthy children, possibly preventing the otherwise inevitable evolution towards clinical development of T1D. With the increasing availability at lower costs of autoantibody screening and HLA genotyping for T1D and the promising ongoing research into powerful calcitriol analogs, secondary and perhaps tertiary prevention of T1D in the general population may be feasible in the near future.

## Figures and Tables

**Figure 1 children-10-00862-f001:**
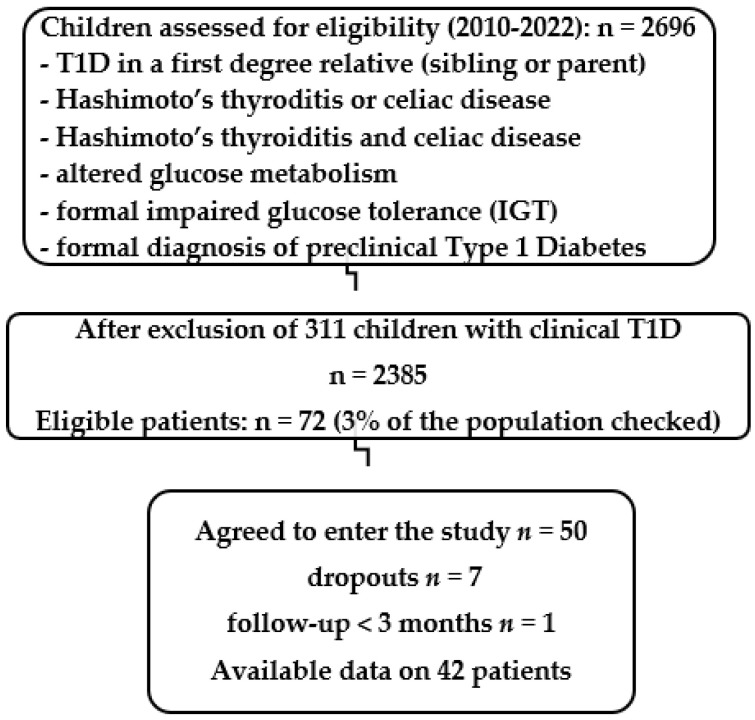
A STARD (STAndards for the Reporting of Diagnostic accuracy studies) flow diagram showing the disposition of subjects participating in the PRECAL study.

## Data Availability

The analytical individual follow-up data for this study is available on request from the corresponding author. All necessary patient data is contained within the article.

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
