# Peer review of "Secondary Prevention of Diabetes Type 1 with Oral Calcitriol and Analogs, the PRECAL Study"

_children, 2023, doi:10.3390/children10050862_

Round 1

Reviewer 1 Report

This is the prospective interventional non-randomized clinical trial, the PRECAL study (ISRCTN17354692), aimed to analyse the effect of calcitriol (0.05 mcg/Kg/day) 0.25-0.5x1-3/day or its synthetic analog, paricalcitol (thrice the calcitriol dose) 1-4 mcg x 1-3/day p.o. for 3-6 months in high risk children 44+T1Ab, 6+HLA, with follow up for 3 years.  The authors concluded that secondary pre- vention of T1D with calcitriol and its analog paricalcitol seems possible and reasonably safe, if started soon enough after seroconversion.

Comments to the Authors:

1.                   The overall language used in the manuscript needs improvement, in order to correct numerous mistakes.

2.                   Line 18 …at ages 2/6…is it ages 2-6 years?

3.                   Please mention the study design in abstract

4.                   Please add the aim of the study in Abstract section and in the manuscript

5.                   The Introduction section is too long, 4 pages, please make it shorter, this is research manuscript and not the chapter in the book or review article

6.                   This is not RCT, but is it possible for authors to make Flow chart in Methodology section related to subject’s inclusion and research design

7.                   Please, add more information in Material and Methods regarding subject’s recruitment (number of FDR, prediabetics, diabetics, non-FDR

8.                   Please add more information about recruitment centres (the level of care)

9.                    Please define risk HLA for inclusion

10.               Conclusion section must be shorter and without discussion

Author Response

Authors’ response to R1

This is the prospective interventional non-randomized clinical trial, the PRECAL study (ISRCTN17354692), aimed to analyse the effect of calcitriol (0.05 mcg/Kg/day) 0.25-0.5x1-3/day or its synthetic analog, paricalcitol (thrice the calcitriol dose) 1-4 mcg x 1-3/day p.o. for 3-6 months in high risk children 44+T1Ab, 6+HLA, with follow up for 3 years.  The authors concluded that secondary pre- vention of T1D with calcitriol and its analog paricalcitol seems possible and reasonably safe, if started soon enough after seroconversion.

Comments to the Authors:

  1. The overall language used in the manuscript needs improvement, in order to correct numerous mistakes.

A: The revised version has now been checked and corrected by a native English speaker.

  1. Line 18 …at ages 2/6…is it ages 2-6 years?

A: No, it is 2 and 6, we have now clarified that point.

  1. Please mention the study design in abstract

A: Added as suggested L21

  1. Please add the aim of the study in Abstract section and in the manuscript

A: We added a small separate section “Aim of the study” in the introduction and a separate phrase in the abstract, as suggested

  1. The Introduction section is too long, 4 pages, please make it shorter, this is research manuscript and not the chapter in the book or review article

A: We have now modified the introduction respecting your useful suggestion.

  1. This is not RCT, but is it possible for authors to make Flow chart in Methodology section related to subject’s inclusion and research design

A: We have now added a Flow chart in Methods Section as suggested.

  1. Please, add more information in Material and Methods regarding subject’s recruitment (number of FDR, prediabetics, diabetics, non-FDR

Α: All this information is presented in detail in Table 1 but we do now also present these numbers in the Methods section as suggested.

  1. Please add more information about recruitment centres (the level of care)

Α: We clarified this as suggested at the beginning of the Methods section

  1. Please define risk HLA for inclusion

A: We added this phrase and reference L85 in the introduction “HLA remains, by far, the strongest predictor of T1D risk [15]” and also added this reference and phrase in the Methods section. “Regarding HLA risk, any high-risk child as defined above that was negative to the available T1Abs and had a predisposing HLA epitope(s) regardless of the specific risk for this genotype was included in the study.”

  1. Conclusion section must be shorter and without discussion

Α: We revised the conclusion as suggested.

Reviewer 2 Report

Dear Editor in Chief,

Thanks for giving me the opportunity to review this interesting manuscript entitled “Secondary PREvention of Diabetes Type 1 with oral CALcitriol  and analogs (the PRECAL study. The article is interesting but has some major design, methodological and results interpretation pitfalls that renders it unsuitable for publication in its current form. Attached below my comments that I hope would add to the manuscript.

Best regards,

Nouran Y Salah, MD

Assistant Professor of pediatric diabetes, endocrinology and metabolism, ASU, AFCM

Comments to the Authors,

I read with great interest the manuscript entitled ” Secondary PREvention of Diabetes Type 1 with oral CALcitriol  and analogs (the PRECAL study. The topic is interesting and promising but has some majir methodological and results pitfalls which renders the results inaccurate. 

Major comments:

-Abstract: The abstract is nit sectioned and lacks clear description of the methodology and results. It would be nice to clarify the methodology and results with clear description of the sample size. . 

-Introduction: The introduction is too long. You can remove the sections “Immunomodulating properties”, Optimal 25(OH) concentrations and vitamin D supplementation roduction”, “1.2.4 The active hormone calcitriol and it’s analog paricalcitol  and summarize the section “A window of opportunity to prevent and revert seroconversion, and intercept the development of multiple T1D autoantibodies”.

-Methodology: 

The age range of the studied participants is too wide o.65-16.37 yrs which greatly affect the validity of the results.

Inclusion criteria:

The population studied is not homogenous and together with the small sample size renders the results invalid. “High-risk children, either positive for at least one T1Ab (IAA, IA2, GAD, 253 ICA, ZnT8 if available), (T1Ab+, n=44) or with predisposing HLA A, DQ, DR haplotypes 254 and genotypes (HLA+, n=6, tested if they were negative for all available T1Abs) were included. These children were screened either because of T1D in a first degree relative, or 256 the diagnosis of Hashimoto’s and/or celiac disease, or altered glucose metabolism: fasting 257 plasma glucose repeatedly > 100 mg/dl, and/or borderline or abnormal HbA1c > 5.7%, or 258 formal impaired glucose tolerance (IGT) at the oral glucose tolerance test (2h-sample > 140 259 mg/dl), or even formal diagnosis of diabetes (verified plasma fasting glucose > 125 mg/dl, 260 or > 200 in the 2h sample at the OGTT). 

Is it justified to withhold insulin therapy from children with  proven T1D. You said you did not administer insulin in these  children. 

 “Inclusion criteria was proven T1D susceptibility regardless of the glucose metabolism status.  Nine had varied impaired glucose tolerance (IGT) (T1Ab+), 4 with pre-T1D (3 T1Ab+ and 265 1 HLA+), and 9 had new-onset T1D (T1Ab+) but not on insulin replacement yet.

Why use activated form of vitamin D in these patients. It is known that people with T1D have no problem in vitamin D activation. 

The duration of follow up of the patients are not clear. Were all patients studied for 6 months only and if so what investigations were repeated and when.

Was ethical committee approval taken and if taken could you please mention its number.

-Results: 

The results are just descriptive with no statistical analyses done.

It would be nice to add comparison of the studied parameters at baseline 3, 6 months and study endpoint. It would be also nice to add percent changes of these variables and correlations and regression analysis of these variables with vitamin D status.

It would have been better if the authors did sample size calculation prior to the study to identify the representative sample size. The small sample size should be added to the limitations 

-Conclusion: The conclusion is too long and discuss the results with references. The conclusion should be short, clear without discussion and references.

Please add a limitations section at the end of the study.

Minor comments:

-Grammatical and language editing are advised by a native English speaker as the study has many grammatical errors. 

-Abstract: 

Please mention any abbreviation in details the first time e.g. line 18 T1D: type 1 diabetes (T1D).

T1Ab…

-Discussion: 

It would be nice to avoid using the term “our study”

Author Response

Response to R2

Comments to the Authors,

I read with great interest the manuscript entitled ” Secondary PREvention of Diabetes Type 1 with oral CALcitriol 2  and analogs (the PRECAL study. The topic is interesting and promising but has some majir methodological and results pitfalls which renders the results inaccurate. 

 Major comments:

-Abstract: The abstract is nit sectioned and lacks clear description of the methodology and results. It would be nice to clarify the methodology and results with clear description of the sample size.

A: We have now reformed our abstract according to your suggestions and the Journal’s requirements “Abstract: The abstract should be a total of about 200 words maximum. The abstract should be a single paragraph and should follow the style of structured abstracts, but without headings”.

  • Study design was added.
  • Aim of the study was added.
  • The methodology and the results within the abstract have been revised as clearly possible – according to your suggestions and respecting as possible the limited length of about 200 words for the abstract, required by “Children”.

\-Introduction: The introduction is too long. You can remove the sections “Immunomodulating properties”, “Optimal 25(OH) concentrations and vitamin D supplementation roduction”, “1.2.4 The active hormone calcitriol and it’s analog paricalcitol”  and summarize the section “A window of opportunity to prevent and revert seroconversion, and intercept the development of multiple T1D autoantibodies”.

A: We have now modified the introduction reducing its size in half respecting all your useful suggestions.

-Methodology: 

The age range of the studied participants is too wide o.65-16.37 yrs which greatly affect the validity of the results.

A: This limitation is now thoroughly discussed as correctly suggested in the limitations paragraph. Despite the wide age range, they all were in a similar autoimmune status regarding T1Abs. The importance of timely prevention of seroconversion is thoroughly discussed throughout the paper. Furthermore, with the statistical analysis we added, age did not seem to have an effect to time needed for T1Abs negativation or the negativation magnitude.

We also highlighted the fact that T1Abs are a useful marker but: “This means that although T1Abs are related to the risk of T1D, they are not the primary mediators and their presence of titters do not directly correlate with the degree of β-cell dysfunction” with the proper reference.

We acknowledge the important limitations in the power of this study, but we hope that with our revised paper after your precious reviewing we are opening the way for a novel secondary prevention strategy, which seems 100% effective in healthy children at risk for T1D, even at a limited sample, if started soon enough after seroconversion. We also hope that our data together with the extensive analysis of the available literature we did, will also drive the medical community and public health authorities to reconsider primary prevention strategies with the development of global and optimal vitamin D supplementation/food fortification programs.

Inclusion criteria:

The population studied is not homogenous and together with the small sample size renders the results invalid. “High-risk children, either positive for at least one T1Ab (IAA, IA2, GAD, 253 ICA, ZnT8 if available), (T1Ab+, n=44) or with predisposing HLA A, DQ, DR haplotypes 254 and genotypes (HLA+, n=6, tested if they were negative for all available T1Abs) were included. These children were screened either because of T1D in a first degree relative, or 256 the diagnosis of Hashimoto’s and/or celiac disease, or altered glucose metabolism: fasting 257 plasma glucose repeatedly > 100 mg/dl, and/or borderline or abnormal HbA1c > 5.7%, or 258 formal impaired glucose tolerance (IGT) at the oral glucose tolerance test (2h-sample > 140 259 mg/dl), or even formal diagnosis of diabetes (verified plasma fasting glucose > 125 mg/dl, 260 or > 200 in the 2h sample at the OGTT). 

Is it justified to withhold insulin therapy from children with  proven T1D. You said you did not administer insulin in these  children.

A: We have now clarified this point. No child was held from insulin replacement therapy at any time point needed. These children were asymptomatic at inclusion but met the criteria for formal diagnosis of T1D, as clarified.

Why use activated form of vitamin D in these patients. It is known that people with T1D have no problem in vitamin D activation.

A: To directly answer this reasonable question we have added this phrase “The rationale of using less calcemic calcitriol analogs as paricalcitol or even clearly non-calcemic calcitriol analogs inducing immunomodulation for health and disease, lies on the ability to use clearly higher pharmaceutical doses than those that could be tolerated by using calcitriol due to its calcemic effects.”

The duration of follow up of the patients are not clear. Were all patients studied for 6 months only and if so, what investigations were repeated and when.

A: No patient except from those reported as dropouts has stopped follow-up. At the time the data of this ongoing prospective study were collected and analyzed (12-year data, 2010-2022) the time to T1Ab negativation as well as the follow-up since negativation was achieved is given for each patient in table 1.

 Was ethical committee approval taken and if taken could you please mention its number.

A: PRECAL is a registered study, and all this information is available online. We did add the ethical approval number as requested.

-Results: 

The results are just descriptive with no statistical analyses done.

It would be nice to add comparison of the studied parameters at baseline 3, 6 months and study endpoint. It would be also nice to add percent changes of these variables and correlations and regression analysis of these variables with vitamin D status.

It would have been better if the authors did sample size calculation prior to the study to identify the representative sample size. The small sample size should be added to the limitations.

A: Thank you for these very constructive remarks of great value. The revised paper has a STARD flow diagram, a sample size calculation, and a formal statistical analysis with a relevant section in Methods and Results with the aid and guidance of two statistics experts, mentioned in the acknowledgements section. The small sample size is added to the limitations as well.

We assessed the T1Ab titers and the magnitude of their negativation as well as time needed to negativation of T1Abs in relation to all recorded parameters before treatment initiation.

Verified T1Ab positivity is what gives susceptibility to development of T1D, but not the level of the T1Ab it-self, as shown by the comparison we mention in the Discussion section between patients presenting with clinical T1D and children included in the PRECAL study, where there was no difference in T1Ab titters. The same stands also other autoimmune diseases such as Hashimoto’s where somebody may have thousands of anti-TPO and/or anti-TG abs but being euthyroid and someone with a mediocre rise in abs can present with subclinical or with severe clinical hypothyroidism.

We showed that there was no statistical difference between vitamin D status as 25(OH)D concentrations at inclusion and that vitamin D – possibly due to the limited number of patients. We also checked and vitamin D at inclusion does not have an impact in time to negativation needed not the magnitude of negativation.

All children included were under the same optimal cholecalciferol supplementation, and more importantly: whereas it would be of interest to compare 1-25(OH)2D3 concentrations, it is known that paricalcitol interferes with 1,25-dihydroxyvitamin D measurement by liquid chromatography–tandem mass spectrometry assays turning unreliable any such attempt. [El-Khoury, J. M., Bicer, F., Bunch, D. R., Yuan, C., & Wang, S. (2013). Does Paricalcitol (Zemplar®) interfere with 1,25-dihydroxyvitamin D measurement by liquid chromatography–tandem mass spectrometry assays? Clinica Chimica Acta, 415, 230–232. doi:10.1016/j.cca.2012.10.050]

-Conclusion: The conclusion is too long and discuss the results with references. The conclusion should be short, clear without discussion and references.

Α: We revised the conclusion as suggested.

Please add a limitations section at the end of the study.

A: Added as suggested and included all the above limitations the reviewer explicitly underlined.

Minor comments:

-Grammatical and language editing are advised by a native English speaker as the study has many grammatical errors.

A: The revised version has now been checked and corrected by a native English speaker.

-Abstract: 

Please mention any abbreviation in details the first time e.g. line 18 T1D: type 1 diabetes (T1D).

T1Ab…

A: Thank you for noticing this. All these were corrected as suggested.

It would be nice to avoid using the term “our study”

A: Thank you for noticing this. All these were changed to “the PRECAL study”

Round 2

Reviewer 1 Report

The authors made changes in the manuscript, according to the suggestions